# Land Use Classification of the Deep Convolutional Neural Network Method Reducing the Loss of Spatial Features

**DOI:** 10.3390/s19122792

**Published:** 2019-06-21

**Authors:** Xuedong Yao, Hui Yang, Yanlan Wu, Penghai Wu, Biao Wang, Xinxin Zhou, Shuai Wang

**Affiliations:** 1School of Resources and Environmental Engineering, Anhui University, Hefei 230601, China; yaoxd9501@163.com (X.Y.); wuph@ahu.edu.cn (P.W.); wangbiao-rs@ahu.edu.cn (B.W.); zxx0331@163.com (X.Z.); 2Institute of Physical Science and Information Technology, Anhui University, Hefei 230601, China; 3Anhui Engineering Research Center for Geographical Information Intelligent Technology, Hefei 230601, China; 4Key Laboratory of Ecological Protection and Restoration of Wetland in Anhui Province, Anhui University, Hefei 230601, China; 5School of Resource and Environmental Science, Wuhan University, Wuhan 430079, China; 2010282050194@whu.edu.cn

**Keywords:** remote sensing, semantic segmentation, DCNNs, coordconv, ISPRS, high resolution

## Abstract

Land use classification is a fundamental task of information extraction from remote sensing imagery. Semantic segmentation based on deep convolutional neural networks (DCNNs) has shown outstanding performance in this task. However, these methods are still affected by the loss of spatial features. In this study, we proposed a new network, called the dense-coordconv network (DCCN), to reduce the loss of spatial features and strengthen object boundaries. In this network, the coordconv module is introduced into the improved DenseNet architecture to improve spatial information by putting coordinate information into feature maps. The proposed DCCN achieved an obvious performance in terms of the public ISPRS (International Society for Photogrammetry and Remote Sensing) 2D semantic labeling benchmark dataset. Compared with the results of other deep convolutional neural networks (U-net, SegNet, Deeplab-V3), the results of the DCCN method improved a lot and the *OA* (overall accuracy) and mean *F1* score reached 89.48% and 86.89%, respectively. This indicates that the DCCN method can effectively reduce the loss of spatial features and improve the accuracy of semantic segmentation in high resolution remote sensing imagery.

## 1. Introduction

Land use classification information offers a significant indication of human activities in an urban environment [1]. This information can provide the basic datasets for change detection [2], landscape pattern [3], and urban heat island effects [4]. With the rapid development of remote sensing technology, this method has become a major way to obtain land use information, and innumerable high resolution remote sensing images are used to extract spatial information regarding urban land use. However, improvements in spatial resolution increase the internal variability of homogenous land cover units and decrease the statistical separability of land cover classes in the spectral space, not necessarily achieving better classification [5]. The internal variability of the high-resolution images makes the land use classification more challenging [6,7].

In land use classification methods, the pixel-based method applies statistical knowledge to extract features based on the spectral characteristics of ground objects—this method is mainly used for low resolution remote sensing images (10–30 m) [8,9]. In contrast, high resolution remote sensing images make the land cover image characteristics more complex and diverse. Pixel-based methods cannot take the context and texture around the pixels into consideration, which generates much of the salt and pepper noise, and these methods cannot obtain object-level information [10]. Because of the limitations of pixel-based classification methods, object-based methods were proposed [11] and obtained a better performance in terms of high-resolution image classification and object identification [12]. The object-based method first separates the spatially and spectrally similar pixels at different scale levels as segmented objects that can effectively identify urban land cover classes [13]. Then, the texture and geometric features of the segmented objects are calculated as a rule set to achieve the final image classification. The object-based method effectively avoids the classification errors caused by spectral differences in pixel-based classification and eliminates the influence of salt and pepper noise. Therefore, object-based modules are also adopted by commercial software systems, such as the eCognition [14], and ENVI [15] (The Enviroment for Visualizing Images) systems. However, this method overlooks the semantic functions or spatial configurations [16] and depends heavily on the selected land-cover classification system and the accuracy of the land-cover classification [17], and the specific object features used in the rule set are not applicable to various ground objects. Thus, advanced methods to automatically extract features from high-resolution remote sensing images are still urgently in demand.

Recently, deep learning has provided an effective method to automatically learn and identify features from large amounts of data, which is the main difference compared to the traditional machine learning methods. With the development of deep learning theory, many deep learning architectures, such as convolutional neural network (CNN) and recurrent neural networks (RNN [18]), have achieved state-of-the-art performance in terms of computer vision [19], object detection [20], and natural language processing.

Jonathan Long et al. [21] used standard convolutional layers to replace the fully connected layer in a CNN and proposed the fully convolutional network (FCN) to promote the large-scale development of image semantic segmentation. The FCN architecture maintains the two-dimensional structure of the feature maps in upsampling, which is quite different from the standard CNN. Recently, as an extension of the FCN, the deep convolutional neural network (DCNN) is rapidly being adopted for image classification [22,23,24,25]. 

In general, FCNs are composed of an encoder part and a decoder part. The encoder part comprises the stacks of “convolutional-pooling” layers, which are used as multi-scale feature extractors and aimed at extracting abstract high-level features. The decoder part contains the upsampling and skip layer. The upsampling is used to recover the input feature map resolution by deconvolution or interpolation and to generate segmentation maps with the same resolution as the original image. However, the deconvolution or interpolation operation only generates coarse segmentation maps, which lack an accurate localization of the object boundaries and high-frequency details. Therefore, the skip layer is used to fuse the previous input features with the output features from the upsampling. However, there is still a loss of detailed information from the reduced feature resolution. To overcome this, tremendous efforts have been made. Saining Xie et al. [26] developed a new algorithm called holistically nested edge detection (HED) to resolve the challenging ambiguity problem in edge and object boundary detection via multi-scale feature learning. Marmanis et al. [27] combined the semantic segmentation with semantically informed edge detection to eliminate the effect of the associated loss of effective spatial resolution washing out the high-frequency details, leading to blurry object boundaries—this method made the boundaries more explicit and improved the overall accuracy. Gong Cheng et al. [28] proposed discriminative CNNs and trained them by optimizing a new discriminative objective function that imposes a metric learning regularization term on the CNN features to address the problem of within-class diversity and between-class similarity. Wei Liu et al. [29] presented a simple technique that introduced the global context to the FCN to enlarge the receptive field and augment the features at each location. This change achieved state-of-the-art performance for semantic segmentation. Furthermore, Chen et al. [30,31,32] discovered that consecutive convolution and pooling operations reduced the feature resolution and caused a loss of spatial detailed information in the downsampling. Therefore, they used atrous convolutional layers to replace some standard convolution and pooling layers. Without introducing extra parameters and changing the feature resolution, the atrous convolutional layer could effectively enlarge the receptive fields, decrease the loss of spatial information, and improve the classification accuracy. However, the consecutive atrous convolutional layers still accounted for the missing spatial information, which is called the “gridding effect”. PanquWa et al. [33] proposed a hybrid dilated convolution (HDC) framework by the superposition of a receptive field in various atrous rates to counter the gridding effect. However, the HDC framework substantially increases the computing costs at the same time. Guangsheng Chen et al. [34] used the DeepLabv3 architecture and an added Augmented Atrous Spatial Pyramid Pool and Fully Connected (FC) Fusion Path layers to tackle the poor classification of small objects and unclear boundaries.

In addition, to solve the coordinate transform problem, which is difficult to achieve in standard CNNs, Liu et al. [35] proposed a significant coordconv module. Because of the lack of the input feature coordinate information, the consecutive convolution operation will discard much of the spatial information at multiple levels. For example, the boundary information in large-scale feature maps will be weakened, and the tiny-scale may vanish. The coordconv module worked by giving the convolution access to obtain its own input coordinates through the use of extra coordinate channels. This module enabled the network to effectively learn the spatial information of different ground objects and eliminated the loss of spatial information, especially the boundary information. The coordconv module achieved a state-of-the-art performance in image classification, object detection, generative modeling, and reinforcement learning.

To further reduce the loss of high-frequency details and object boundaries in high resolution remote sensing images, inspired by coordinate convolution, we extended the coordconv module into the FC-DenseNet and designed a novel encoder–decoder architecture called the dense-coordconv network (DCCN) to solve the complex urban land use classification tasks by using high-resolution remote sensing images. The study of [36] demonstrated that the DenseNet could alleviate the vanishing gradient problem, encourage feature reuse, and effectively reduce the number of parameters; thus, DenseNet is considered the best framework for semantic segmentation [37]. The FC-DenseNet, which is an extension of DenseNet that works as an FCN and has the same advantages mentioned above, has achieved an excellent performance in image semantic segmentation. Therefore, the FC-DenseNet is used as the overall structure of our network. In addition, to analyze the effect of coordconv module on network, the paper discusses the performance of the coordconv module located in different level feature maps.

The remainder of this paper is arranged as follows. Section 2 mainly introduces the network architecture and coordconv module. Section 3 presents the experimental results, and Section 4 introduces the experimental discussion. The conclusion is shown in Section 5.

## 2. Methods

### 2.1. Network Architecture

Similar to other encoder–decoder architectures, this paper also employed an encoder–decoder architecture based on the architecture of the FC-DenseNet. The coordconv layer in the top of network was used to obtain its own input coordinate information and determine the pixel positions in the feature maps. In the encoder part, a series of dense blocks were applied to extract the abstract features, and there was a transition layer between the dense blocks. In the decoder part, the low-level features and the high-level features from the transposed layers were directly concatenated by a skip connection to form a new dense block input. At the end of the designed network, the segmentation maps were outputted after a 1 × 1 convolution. The complete architecture of our network is shown in Figure 1.

As an essential architecture of the proposed network, the DenseNet has a better performance in extracting features than other networks. Generally, the traditional convolutional networks with *L* layers will produce *L* connections, while the DenseNet has *L* × (*L* + 1)/2 connections. For each layer, the feature maps of all the preceding layers will be used as the inputs, and its own feature maps are also used as the inputs of the subsequent layers. This dense connectivity has several advantages, such as strengthening feature reuse and reducing the number of parameters, which makes the network more easily trained. In the layers, the nonlinear transformation function of layer *L* is *F_L_*, which was proposed to further improve the information flow between the layers, and the output is *X_L_*. Therefore, the transformation of each layer can be defined as follows:(1)$$X_{L}=F_{L}([X_{0},X_{1},\ldots ,X_{L-1}]),$$
where *X_L_* represents the received total feature maps in layer *L*, and [*X*_0_, *X*_1_, …, *X*_*L*−1_] represents the concatenation of all the preceding feature maps before layer *L*. The nonlinear transformation *F_L_* is often defined as a composite function containing a batch normalization (BN) layer [38], a rectified linear unit (ReLU) [39], and a 3 × 3 convolution layer. Moreover, to suppress the redundancy of the feature maps, the DenseNet uses a growth rate to limit the number of feature maps at each dense block.

### 2.2. The Coordconv Module

The coordconv layer proposed in [35] can be considered an extension of the traditional convolutional layer, as shown in Figure 2. The coordinate information in the feature maps is extracted first and filled in the extra channels, and then it will be concatenated with the original feature maps. After that, the standard convolutional operation is adopted. Generally, the channels are prepared for *i* coordinates and *j* coordinates, and there is a relevant linear transform of both *i* and *j* to normalize them in the range [−1, 1]. The two coordinates are sufficient to record the spatial information in the input feature maps. If necessary, the other channel, which is the *r* coordinate, can also be introduced to deal with specific cases. The *r* coordinate can be calculated as follows:(2)$$r=\sqrt{{\left(i-h/2\right)}^{2}+{\left(j-w/2\right)}^{2}},$$
where *h* and *w* are the sizes of the feature maps. The *i* and *j* represent the basic coordinates.

Compared with the standard convolutional layer, the coordconv layer can be considered a spatial attention unit. The coordconv layer stores the horizontal and vertical pixel information of the object boundary with extra channels as the input. Therefore, the proposed network could output the feature maps with rich information after consecutive convolutional operations. This coordconv module is effective in highlighting the detailed information and decreasing the loss of features, as well as benefiting the pixel-level segmentation, which is quite different from the standard convolutional layer. Figure 3 shows the characteristics of both the coordconv layer and convolutional layer. As seen from Figure 3, the coordconv layer was feasible to strengthen the boundary information and decrease the internal variability in each class.

### 2.3. Implementation Details of the Networks

In this experiment, the DCCN consisted of ten dense blocks, four transition layers, and five transposed layers in the encoder and decoder parts. The detailed architecture is shown in Figure 4. In the top of the network, we designed a coordconv layer to obtain the spatial information in feature maps. Before applying the first dense block, a 7 × 7 convolutional layer with stride = 2 was used to extract the features as the initial input. In the encoder part, the transition layer in the adjacent dense blocks mainly contained a 1 × 1 convolutional layer, a dropout layer, and a 2 × 2 average pooling layer with stride = 2. In the decoder part, there was a 3 × 3 transposed convolutional layer, with stride = 2 and a skip connection layer in each dense block (except the last block). The skip connection layer included a 3 × 3 convolutional layer. At the end of the network, a 1 × 1 convolutional layer with a ReLU activation function was used to output the final feature maps.

## 3. Experiments

### 3.1. Dataset

In this research, we trained the proposed model on two images datasets from the ISPRS 2D semantic labeling contest, which were open benchmark datasets that are available online. The two datasets were both comprised of very high-resolution images from two cities in Germany (Potsdam and Vaihingen). These datasets were provided to perform the semantic labeling, and they are generally classified into six common land cover classes: buildings, impervious surfaces, low vegetation, trees, cars, and clutter or background.

#### 3.1.1. ISPRS Potsdam Dataset

The Potsdam dataset in total contained 38 high-resolution true orthophoto (TOP) patches, which were extracted from a larger true orthophoto (TOP) mosaic. Each patch contained approximately 6000 × 6000 pixels at a spatial resolution of 5 cm. However, only 24 patches were labeled with the pixel-level ground truth, and the remaining 14 patches were unreleased. These labeled images consisted of infrared-red-green-blue multispectral images, however, we only used the infrared-red-green (IRRG) channels of the multispectral images as samples. Furthermore, to reduce the interference from clutter or background and train the model more effectively, we selected five labeled images that did not include a large amount of typical background, such as water, as the validation set, and the other labeled images were used as training samples.

#### 3.1.2. ISPRS Vaihingen Dataset

The Vaihingen dataset contained 33 high resolution patches extracted from a larger true orthophoto (TOP) mosaic. Each patch contained approximately 2000 × 2000 pixels at a spatial resolution of 9 cm. However, only 16 patches had a public ground truth. These labeled images only consisted of the infrared-red-green (IRRG) images. In our research, we randomly selected three patches as the validation set, and the others were used as training samples.

### 3.2. Experimental Setup

#### 3.2.1. Implementation Details

Considering the number of training samples and the diversity of the samples, we split the labeled images with 6000 × 6000 pixels into three scales: 256 × 256, 320 × 320, and 448 × 448. Therefore, we had a total of 21,527 subset images for training our model. Due to the limited memory of the GPU, we split the size of testing data into 3000 × 3000 pixels.

For more suitable network training, we set an automatic learning rate controlled by the training epochs. There was a total of 200 epochs in the training process, and the batch size was 10. The initial learning rate was set at 0.001. When the number of epochs reached 20, the learning rate was 0.0001. Similarly, when the number of epochs reached 100 and 150, the learning rate became 0.00005 and 0.00001, respectively. The computational cost of training the model was approximately 90 h, and the average testing time of one image (3000 × 3000) was approximately 2 s. In addition, the Adam is an adaptive optimizer with high computational efficiency and low memory requirement, so we adopted the Adam algorithm as the optimizer to optimize the network and update all the parameters.

#### 3.2.2. Evaluation Metrics

According to the benchmark rules, the overall accuracy (*OA*) and *F1* score are always applied to evaluate the quantitative performance. The *OA* is calculated as follows:(3)$$OA=\sum_{i=0}^{k}\frac{p_{i}}{N},$$
where *k* represents the classes, *P_i_* is the total number of correct pixels in class *i*, and *N* represents the number of image pixels. The *F1* score can be calculated through the precision and recall metrics, and it is a powerful evaluation metric for the harmonic mean of the precision and recall metrics. The *F1* score can be calculated by:(4)$$F1=2\times \frac{precision\times recall}{precision+recall},$$
where
(5)$$recall=\frac{TP}{TP+FN},precision=\frac{TP}{TP+FP}.$$

The recall represents the proportion of correct pixels in the ground truth, and the precision is the proportion of correct pixels in the prediction result. *TP*, *FP*, and *FN* represent the true positives, false positives, and false negatives, respectively. These values can be calculated using the pixel-based confusion matrix per patch.

In addition, another metric, *IoU*,(the intersection over union) was adopted to evaluate the shape and area of all classes. The *IoU* is the intersection of the prediction and ground truth regions over their union for a specific class. The mean *IoU* can be calculated by averaging the *IoU* of all classes:(6)$$IoU=\frac{TP}{FN+TP+FP}.$$

The numerator and denominator all represent the areas of detected objects.

### 3.3. Experimental Results

All the prediction results for the five datasets are displayed in next section (DCCN). We can see that each image has a better result and that the six land cover classes are classified completely and are very close to the ground truth images. To quantitatively demonstrate the excellent performance, the average *OA* and mean *F1* score of five images is listed in Table 1. For all the validation images, the overall accuracy (*OA*) and mean *F1* score are 89.48% and 86.89%, respectively. For each class, the accuracies of classifying buildings, impervious surfaces, and cars all exceed 90%, and the accuracies of the buildings are even up to 95.59%. Although the accuracies of classifying trees and low vegetation are lower than those of the other classes, each image still achieves a remarkable accuracy. In a word, all the results show that the proposed DCCN can perform well in semantic segmentation for very high-resolution remote sensing images.

To evaluate the shape and area of detected objects effectively, the *IoU* evaluation results are given in Table 2. The accuracy of the *IoU* results is lower than the pixel-based results. This indicates that there is some difference in various metrics and the *IoU* matric may be more appropriate for the assessment on the shape and area of detected objects.

In addition, we also evaluated the results of the prediction for five validation datasets measured in terms of confusion matrix. As we can see from the Figure 5, the confusion matrix of each image has a better performance. The Prod Accuracy (PA) and User Accuracy (UA) have a higher results and the kappa coefficient of each image exceeds 80%. However, the low vegetation class in the PA of each confusion matrix is worse than other classes. This is quite consistent with the case in Table 1.

## 4. Evaluation and Discussion

### 4.1. The Advancement of the DCCN

To prove the advancement of the proposed DCCN in terms of network architecture and capability, we compared the DCCN with several typical semantic segmentation methods, including the U-net, Deeplab-v3, and SegNet methods, with two datasets. These methods have played a crucial role in the development of semantic segmentation. Many deep convolutional neural networks for classification are extensions of these typical methods, such as those of [40,41]. We implemented and tested these methods on the same experimental dataset.

#### 4.1.1. Testing the Potsdam Dataset

All the testing results on the Potsdam dataset are shown in Figure 6. Due to the coordinate convolution module and the efficient feature extraction of multilevel features, the segmentation results of the proposed DCCN are highly complete and precise, while the U-net, Deeplab-V3, and SegNet methods produced serious misclassifications and missing pixels, especially in terms of the impervious surfaces and buildings (see the pink boxes in Figure 6). This suggests that the coordconv mechanism and efficient dense block architecture are significant in improving accuracy. Furthermore, to quantify the results among the proposed DCCN and the U-net, Deeplab-V3, and SegNet methods, the overall accuracy (*OA*) and mean *F1* score of the five classes are listed in Table 3. We can see that the DCCN obviously outperforms the other mainstream models in terms of both the *OA* and mean *F1* score. The *OA* and mean *F1* score of the DCCN are 1.36% and 1.16% higher, respectively, than those of the SegNet model and far exceed those of the Deeplab-V3 and U-net methods. However, in terms of the individual classes, the accuracy in the low vegetation class for the DCCN is 78.26%, which is 1.51% lower than that of SegNet. Although there are a few shortcomings in a specific class, the proposed DCCN still achieves a better performance for the classification of very high-resolution images. Similarly, the *IoU* results between the DCCN and mainstream models are displayed in Table 4. The *IoU* result of DCCN is obviously better than other three networks. It indicates that the proposed DCCN is more effective for assessing the shape and area of detected objects. Although the *IoU* metrics are different from the pixel-based system, the DCCN results of these metrics all achieve high scores. This shows that the proposed DCCN is a successful method.

#### 4.1.2. Testing the Vaihingen Dataset

Our proposed model achieved a superior performance on the previous dataset. To prove the advancement of the proposed DCCN further, we transferred the learning of these networks to another ISPRS benchmark (Vaihingen) dataset, described in detail in Section 3, and the testing results of all networks are displayed in Figure 7. In these testing images, the results of the DCCN show a better performance in terms of the completeness of an object boundary than any of the other methods, as seen in Figure 7e. However, these prediction results are still seriously affected by the limited number of labeled samples (approximately 1300 256 × 256 samples in total). Therefore, there are a few false classifications and missing pixels in the different prediction results. Additionally, the overall accuracy and *F1* score of all the methods are listed in Table 5 and the *IoU* results are shown in Table 6. In spite of the limited number of samples, the DCCN still achieved a higher accuracy than the other typical methods, in terms of both the *OA* and *F1* score and *IoU* in each class. This shows that the proposed DCCN is a successful and powerful method for urban land use classification with a small sample size.

### 4.2. Effect of the Coordconv Module

To properly highlight the advantage of adding the coordconv layer to the network, we used the results of the DCCN without the coordconv layer as the baseline, which is equivalent to the improved FC-DenseNet (see Figure 8a). The proposed model with one coordconv layer (DCCN) outperformed the baseline model with improvements of 0.27% and 0.12% in the *OA* and mean *F1* score, respectively, as shown in Table 7. This showed that the coordconv module had an effective positive impact on feature extraction. Therefore, we attempted to add two coordconv layers to the network, which were located at the top and bottom of the network, called D2CCN (see in Figure 8c). However, the results of the D2CCN were worse than the results of both the baseline model and the DCCN. This indicates that the coordconv layer was useful but may result in negative effects if improperly implemented. Moreover, we also evaluated the kappa coefficient results of the baseline, D2CCN, and DCCN, given in Table 8. Although the kappa coefficients of image 1 and image 2 are lower than the baseline and D2CCN, the Mean kappa of DCCN is the best and reaches 84.80%.

### 4.3. Problem Analysis

Although the proposed DCCN has achieved a satisfying result on land use classification, it should be noted that the accuracy for high resolution images is still affected by some factors, as demonstrated in Figure 9.

Figure 9a–c shows that the vegetation covering on a building’s roof is classified into vegetation class rather than the labeled building class. We can see that the appearance of the vegetation is the same as the building’s roof in Figure 9a. Therefore, the proposed method easily leads to misclassification due to not considering the surrounding context. This type of misclassification causes a negative effect on improving accuracy. Another problem is improperly labeled ground truth images, as shown in Figure 9d–f. A ground truth image is a benchmark for evaluating models. However, many ground truth images are obviously incorrectly labeled as ground objects. The building’s roof and some trees are incorrectly labeled in the background and vegetation classes in the ground truth dataset (Figure 9f), while the prediction result normally performs well in prediction (Figure 9e). This difference between the original image and the ground truth image truly decreases the overall accuracy.

Many improved deep convolutional neural networks have provided some methods that can be used as effective techniques to deal with these problems and to improve accuracy. Recently, much research on attention mechanisms has focused on segmentation and has achieved state-of-the-art performance [42,43,44]. The attention mechanism mainly consists of the local attention module and global attention module. Yongyang Xu et al. [45] designed a local attention unit to fuse different scale feature maps and designed a global attention unit to eliminate the effects on shadow and object covering, which effectively improved the accuracy of road extraction. Moreover, a useful method is to use the entropy to weight the uncertainty of pixels and guide the low-level feature maps, and this method was used by Hongzhen Wang et al. [46] for land use classification. Most importantly, preprocessing and postprocessing of the datasets are also essential to improve accuracy. The cases of buildings being often misclassified as roads may diminish if we add a digital surface model (DSM) to enhance the building features. These methods make a contribution to improving the performance, and we will attempt to add them into our future work.

## 5. Conclusions

In this work, a novel deep convolutional neural network, DCCN, was proposed for semantic segmentation in high-resolution remote sensing images. The major contributions of our method were the proposed encoder–decoder architecture and the introduction of dense blocks as the feature extractors. At the same time, we also added the coordinate convolution module in the network, which obviously improved the *OA* and *F1* score (89.48% and 86.89% in Potsdam, 85.31% and 81.36% in Vaihingen). The proposed DCCN aimed to make full use of multilevel features and eliminate the loss of spatial information. Experiments were carried out on the ISPRS dataset. Six land cover classes were extracted successfully with the proposed DCCN, and the results demonstrated the effectiveness and feasibility of the DCCN in improving the performance for land use classification. The proposed DCCN was compared with other typical networks for semantic segmentation, such as the U-net, SegNet, and Deeplab-V3 models. The experimental results showed that the proposed model performed better than other networks. The *OA* and mean *F1* score of proposed model are 1.36% and 1.16% higher than SegNet method in Potsdam, and far exceed the Deeplab-V3 and the U-net. However, the performance was still affected by complex land covers. In fact, our proposed model has the potential to perform better if some widely used methods, such as the attention mechanism or data preprocessing and postprocessing, are considered.

## Figures and Tables

**Figure 1 sensors-19-02792-f001:**
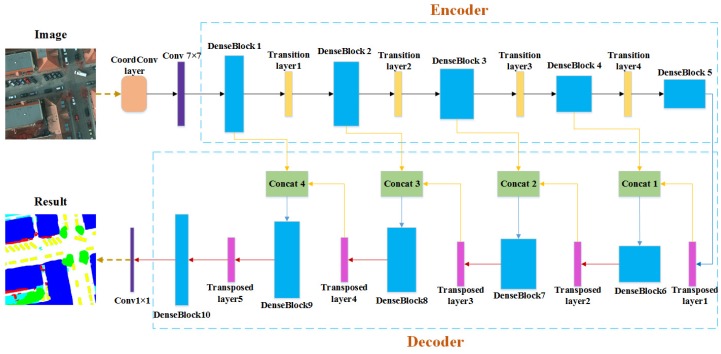
The overall architecture of the dense-coordconv network.

**Figure 2 sensors-19-02792-f002:**
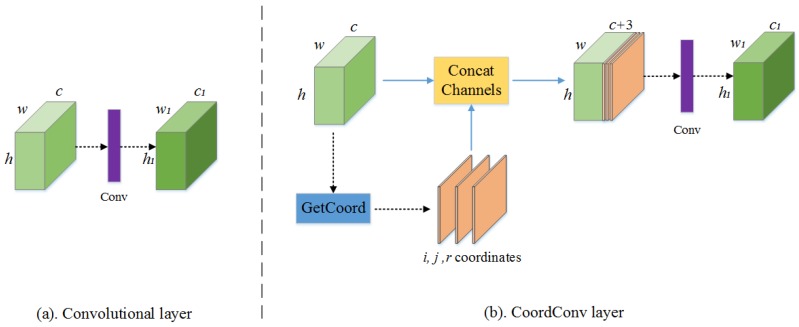
Comparison of the convolutional layer and coordconv layer. (**a**) A standard convolutional layer, where the input feature maps with shape *h* × *w* × *c* to new feature maps with shape *h*_1_ × *w*_1_ × *c*_1_. (**b**) A coordconv layer has the same input shape as the convolutional layer but first concatenates with extra channels from the incoming feature maps. Then, a standard convolutional layer is used.

**Figure 3 sensors-19-02792-f003:**
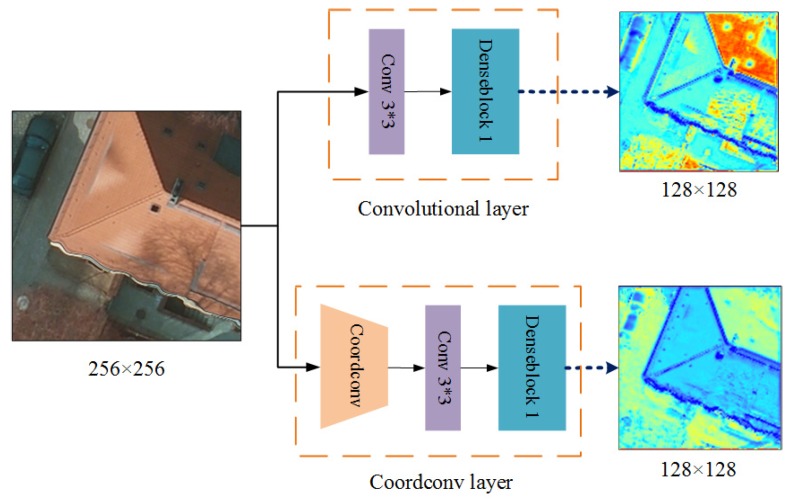
Comparison results between the convolutional and coordconv layers. The up route is a standard convolution including a 3 × 3 convolutional layer and a dense block. The down route is the coordconv layer including a coordinate convolution, a 3 × 3 convolutional layer and a dense block. The dense output maps are all 128 × 128.

**Figure 4 sensors-19-02792-f004:**
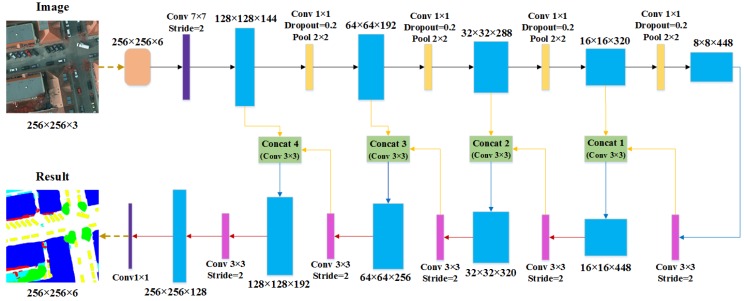
The implementation details of the dense-coordconv network.

**Figure 5 sensors-19-02792-f005:**
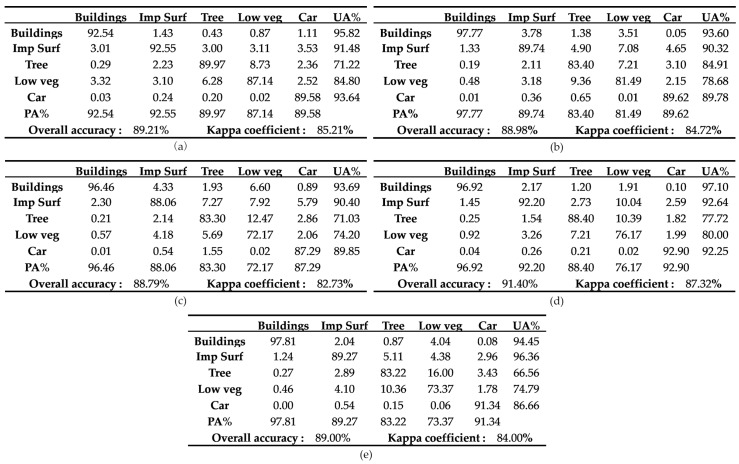
The confusion matrix of five validation images. (**a**–**e**) Confusion matrixes in five classes are calculated by the prediction results and ground truth images. The evaluation metrics in each subfigure contain producer accuracy (PA), user accuracy (UA), overall accuracy (OA) and kappa coefficient. The results in the confusion matrix represent the percentage.

**Figure 6 sensors-19-02792-f006:**
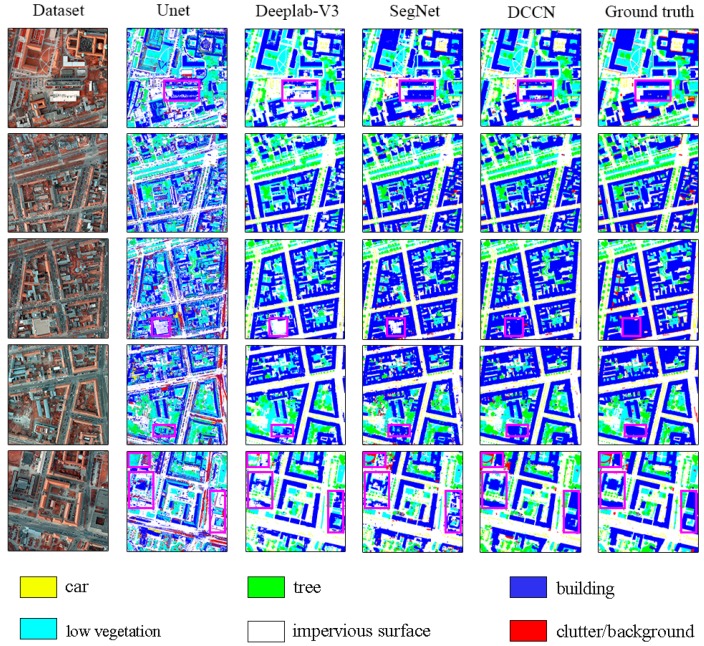
The prediction results of U-net, Deeplab-v3, SegNet and the proposed dense-coordconv network (DCCN) on the validation images.

**Figure 7 sensors-19-02792-f007:**
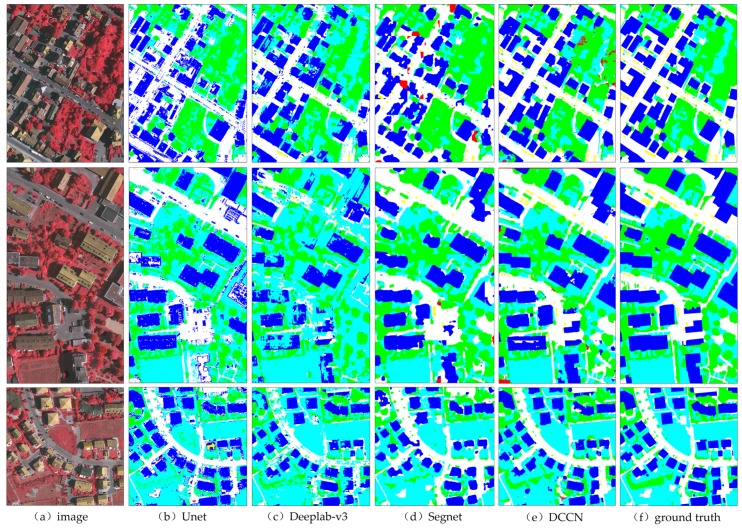
The prediction results of all methods with the Vaihingen dataset.

**Figure 8 sensors-19-02792-f008:**
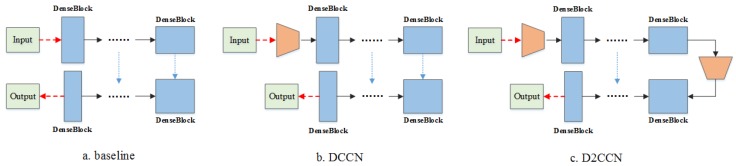
Different strategies of implementing the coordconv module in the network.

**Figure 9 sensors-19-02792-f009:**
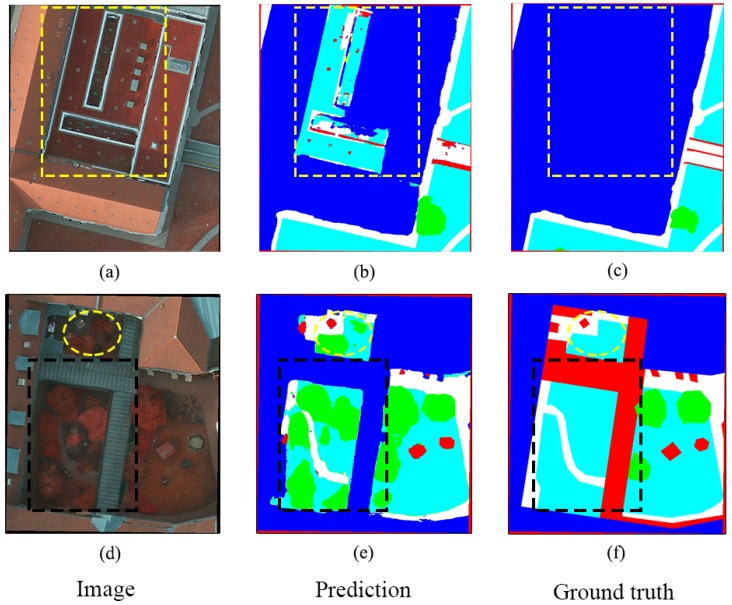
Some factors caused misclassification. (**a**–**c**) A special testing sample that lawn on building roof is classified as low vegetation class shown in (**b**), while the ground truth image (**c**) labels it as building. (**d**–**f**) The building and trees are incorrectly labeled as the background and low vegetation class shown in (**f**), while the (**e**) achieves a correct prediction result.

**Table 1 sensors-19-02792-t001:** The average overall accuracy (*OA*) and mean *F1* score on all the validation datasets and individual datasets.

Validation Datasets	Buildings	Imp Surf	Tree	Low_Veg	Car	Overall Accuracy	Mean *F1* Score
All images	95.59%	91.27%	79.04%	78.26%	90.27%	89.48%	86.89%
image1	94.15%	92.01%	79.50%	85.95%	91.57%	89.21%	88.64%
image2	95.64%	90.03%	84.15%	80.06%	89.70%	88.98%	87.92%
image3	95.05%	89.21%	76.68%	73.17%	88.55%	88.79%	84.53%
image4	97.01%	92.42%	82.72%	78.04%	92.57%	91.40%	88.55%
image5	96.10%	92.68%	73.96%	74.07%	88.94%	89.00%	85.15%

**Table 2 sensors-19-02792-t002:** The results of *IoU* and Mean *IoU* on all the validation datasets.

	*IoU* (%)					Mean *IoU* (%)
	Buildings	Imp Surf	Tree	Low_Veg	Car	
all images	91.57	83.98	66.00	64.52	82.30	77.67
image1	88.95	85.21	65.98	75.37	84.44	79.99
image2	91.65	81.87	72.63	66.75	81.33	78.85
image3	90.57	80.53	62.18	57.69	79.46	74.09
image4	94.18	85.91	70.52	63.99	86.17	80.15
image5	92.50	86.36	58.68	58.82	80.08	75.29

**Table 3 sensors-19-02792-t003:** Comparisons between the proposed DCCN and several typical models.

Method	Building	Imp Surf	Tree	Low Veg	Car	Overall Accuracy	Mean *F1* Score
U-net	62.10%	66.65%	36.30%	57.66%	43.95%	61.28%	53.33%
Deeplab-v3	89.51%	85.83%	72.93%	74.34%	80.54%	83.85%	80.63%
SegNet	93.04%	89.99%	78.98%	79.77%	86.88%	88.12%	85.73%
DCCN	95.59%	91.27%	79.04%	78.26%	90.27%	89.48%	86.89%

**Table 4 sensors-19-02792-t004:** *IoU* results between the proposed DCCN and several typical models.

	*IoU* (%)					Mean *IoU* (%)
	Buildings	Imp Surf	Tree	Low veg	Car	
Unet	45.17	56.46	22.43	41.08	28.19	38.67
DeeplabV3	81.11	75.24	57.58	59.57	67.47	68.19
SegNet	87.12	81.83	65.59	66.69	76.82	75.61
DCCN	91.57	83.98	66.00	64.52	82.30	77.67

**Table 5 sensors-19-02792-t005:** Quantitative comparisons of different methods with the Vaihingen dataset.

Method	Building	Imp Surf	Tree	Low Veg	Car	Overall Accuracy	Mean *F1* Score
U-net	79.95%	79.05%	70.50%	68.10%	15.72%	75.60%	62.66%
Deeplab-v3	83.74%	66.18%	78.78%	64.52%	36.31%	73.86%	65.91%
SegNet	85.97%	80.21%	79.24%	70.36%	26.72%	79.87%	68.50%
DCCN	92.07%	86.43%	82.54%	79.36%	66.40%	85.31%	81.36%

**Table 6 sensors-19-02792-t006:** *IoU* results of different methods with the Vaihingen dataset.

	*IoU* (%)					Mean *IoU* (%)
	Buildings	Imp Surf	Tree	Low veg	Car	
Unet	66.61	65.39	54.94	52.60	14.78	50.86
DeeplabV3	72.64	51.32	65.13	48.99	22.27	52.07
SegNet	74.47	68.95	76.23	40.84	14.61	55.02
DCCN	85.39	76.37	70.80	65.87	50.29	69.74

**Table 7 sensors-19-02792-t007:** The quantitative results of the baseline, D2CCN, and DCCN models with the validation datasets.

Method	Building	Imp Surf	Tree	Low Veg	Car	Overall Accuracy	Mean *F1* Score
baseline	95.51%	91.10%	79.03%	77.94%	90.27%	89.21%	86.77%
D2CCN	94.60%	90.66%	79.53%	77.19%	90.01%	88.81%	86.40%
DCCN	95.59%	91.27%	79.04%	78.26%	90.27%	89.48%	86.89%

**Table 8 sensors-19-02792-t008:** The kappa coefficient results of the baseline, D2CCN, and DCCN.

Method	Image 1	Image 2	Image 3	Image 4	Image 5	Mean Kappa
baseline	84.52%	84.80%	82.25%	87.08%	83.57%	84.44%
D2CCN	85.32%	83.63%	80.39%	87.17%	82.31%	83.79%
DCCN	85.21%	84.72%	82.73%	87.32%	84.00%	84.80%

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
