# Peer review of "Land Use Classification of the Deep Convolutional Neural Network Method Reducing the Loss of Spatial Features"

_sensors, 2019, doi:10.3390/s19122792_

Round 1

Reviewer 1 Report

This paper proposed to add a CoordConv layer to the FC-DenseNet to decrease the loss of spatial information in FC-DenseNet (and similar nets). The paper is well written and easy to follow. The only concern I have is that I believe the contribution of this paper is not enough to be published in this journal (Using a pre-defined layer and add it to a pre-defined network), although it improves the accuracy of the detection. Another point in the accuracy assessment that comes to my mind is that although other papers have used OA, F1 score, and Precision and recall for the assessment, these metrics are not able to show if the shape of the detected object is improved or not. As a suggestion, you can use other metrics that consider the outline of the detected object and evaluate the accuracy. 

There are some minor format problems that should be corrected.

Author Response

Response to Reviewer 1 Comments

  Thank you for your comments, which are all valuable and helpful to revise and improve our paper. Our detailed responses to these comments are listed below.

Point 1: This paper proposed to add a CoordConv layer to the FC-DenseNet to decrease the loss of spatial information in FC-DenseNet (and similar nets). The paper is well written and easy to follow. The only concern I have is that I believe the contribution of this paper is not enough to be published in this journal (Using a pre-defined layer and add it to a pre-defined network), although it improves the accuracy of the detection. Another point in the accuracy assessment that comes to my mind is that although other papers have used OA, F1 score, and Precision and recall for the assessment, these metrics are not able to show if the shape of the detected object is improved or not. As a suggestion, you can use other metrics that consider the outline of the detected object and evaluate the accuracy. 

There are some minor format problems that should be corrected.

Response 1:

        Many deep convolutional neural networks have shown outstanding performance on semantic segmentation. However these methods still have difficulty in reducing the loss of spatial information and strengthening features. The DenseNet can improve features reuse and strengthen features extraction, and the coordconv module can highlight the detailed information and decrease the loss of feature. So this paper proposed a novel model combining the DenseNet with a coordconv module to reduce the loss of spatial information and strengthen features. In addition, to analyse the effect of coordconv module on network further, the paper has discussed the performance of coordconv module located in different-level feature maps. The revised content is in lines 126-128.

 As for the accuracy assessment, your opinion is important. Based on your suggestions, we have attempted to use the IoU metric to measure the area of detected objects. IoU is the intersection of the prediction and ground truth regions over their union for a specific class. This metric may be appropriate to assess the shape of detected object. The revised content is in lines 257-261, 273-276, 311-315. The IoU results of each method are as follows.

Table 2. The accuracy of IoU and mean IoU on all the validation datasets.

IoU(%)

Mean

IoU(%)

Buildings

Imp Surf

Tree

Low_veg

Car

all   images

91.57

83.98

66.00

64.52

82.30

77.67

image1

88.95

85.21

65.98

75.37

84.44

79.99

image2

91.65

81.87

72.63

66.75

81.33

78.85

image3

90.57

80.53

62.18

57.69

79.46

74.09

image4

94.18

85.91

70.52

63.99

86.17

80.15

image5

92.50

86.36

58.68

58.82

80.08

75.29

Table 4. IoU results between the proposed DCCN and several typical models

IoU(%)

Mean

IoU(%)

Buildings

Imp Surf

Tree

Low veg

Car

Unet

45.17

56.46

22.43

41.08

28.19

38.67

DeeplabV3

81.11

75.24

57.58

59.57

67.47

68.19

SegNet

87.12

81.83

65.59

66.69

76.82

75.61

DCCN

91.57

83.98

66.00

64.52

82.30

77.67

Table 6. IoU results of different methods with the Vaihingen dataset

IoU(%)

Mean

IoU(%)

Buildings

Imp Surf

Tree

Low veg

Car

Unet

66.61

65.39

54.94

52.60

14.78

50.86

DeeplabV3

72.64

51.32

65.13

48.99

22.27

52.07

SegNet

74.47

68.95

76.23

40.84

14.61

55.02

DCCN

85.39

76.37

70.80

65.87

50.29

69.74

In addition, we have revised the problem of this manuscript format.

Reviewer 2 Report

The article proposes a new type of neural network, called dense-coordconv network (DCCN), to reduce the loss of spatial features and strengthen object boundaries for land use classification.

The paper is well-written and technically correct. My suggestions and comments for authors to improve the article are given as follows:

1.       Introduction: recent important references directly related to your research is missing. Discuss and cite the following articles:

·         Chen, et al. Fully Convolutional Neural Network with Augmented Atrous Spatial Pyramid Pool and Fully Connected Fusion Path for High Resolution Remote Sensing Image Segmentation. Appl. Sci. 2019, 9, 1816.

·         Cheng, et al. (2018). When deep learning meets metric learning: Remote sensing image scene classification via learning discriminative CNNs. IEEE Transactions on Geoscience and Remote Sensing, 56(5), 2811-2821. doi:10.1109/TGRS.2017.2783902

·         Wei, et al. (2019). Study on remote sensing image vegetation classification method based on decision tree classifier. 2018 IEEE Symposium Series on Computational Intelligence, SSCI 2018, 2292-2297. doi:10.1109/SSCI.2018.8628721

·         Capizzi, et al. (2016). A clustering based system for automated oil spill detection by satellite remote sensing. In Artificial Intelligence and Soft Computing (pp. 613–623). Springer International Publishing doi:10.1007/978-3-319-39384-1_54

2.       Eq. (1): define and describe function FL in more detail.

3.       L. 231: “Adam algorithm” -> reference needed. Why did you select Adam optimizer for this task?

4.       L. 260: replace section name with “Evaluation and Discussion”.

5.       L. 282: “an astonishing performance” -> suggest to avoid such characterization

6.       3.3. Experimental results: provide also the confusion matrix for the classification results.

7.       Tables 2 and 3: supplement the results with ranking based comparison (provide mean rank of each method).

8.       Table 4: the improvement of your method DCCN over baseline is very small. Is it statistically significant? Perform statistical analysis over results to confirm or deny hypothesis about the statistical significance of improvement.

9.       Abstract: “significant improvement” -> a statistical proof is required or avoid using the word “significant”.

10.   L. 322-328: there is no Figure 9 in the article. Should be Figure 8.

11.   Conclusions: rewrite, be more specific, provide main numerical results (such as accuracy) from the proposed network. “the proposed model … was more robust” -> how did you evaluate robustness? Is it robustness to noise? I did not found robustness analysis in the article.

Author Response

Response to Reviewer 2 Comments

Many thanks for your comments and suggestions. We greatly appreciate your comments, which have helped us to improve the quality of the manuscript. We have made changes based on your comments and hope that they will meet with your approval. Our detailed responses to these comments are listed below.

Point 1: Introduction: recent important references directly related to your research is missing. Discuss and cite the following articles:

·         Chen, et al. Fully Convolutional Neural Network with Augmented Atrous Spatial Pyramid Pool and Fully Connected Fusion Path for High Resolution Remote Sensing Image Segmentation. Appl. Sci. 2019, 9, 1816.

·         Cheng, et al. (2018). When deep learning meets metric learning: Remote sensing image scene classification via learning discriminative CNNs. IEEE Transactions on Geoscience and Remote Sensing, 56(5), 2811-2821. doi:10.1109/TGRS.2017.2783902

·         Wei, et al. (2019). Study on remote sensing image vegetation classification method based on decision tree classifier. 2018 IEEE Symposium Series on Computational Intelligence, SSCI 2018, 2292-2297. doi:10.1109/SSCI.2018.8628721

·         Capizzi, et al. (2016). A clustering based system for automated oil spill detection by satellite remote sensing. In Artificial Intelligence and Soft Computing (pp. 613–623). Springer International Publishing doi:10.1007/978-3-319-39384-1_54

Response 1:

Thank you for providing these articles. We have discussed these articles in the Introduction of this manuscript based on your suggestions. These articles are cited in lines 104-106, 89-92, 44-45, 51 of the revised manuscript, respectively.

Point 2: Eq. (1): define and describe function FL in more detail.

Response 2:

Thank you for your comments. Based on your comments, we have describe the FL in detail. The revised contents are located in lines 151-152, 156-158. The revised content is as follow:

“The nonlinear transformation function of layer L is FL, which is proposed to improve the information flow between the layers. The nonlinear transformation FL is often defined as a composite function containing a batch normalization (BN) layer, a rectified linear unit (ReLU) and a 3×3 convolution layer.”

Point 3: L. 231: “Adam algorithm” -> reference needed. Why did you select Adam optimizer for this task?

Response 3:

       Thank you for this suggestion. The Adam optimizer has several advantages such as simple implementation, high computational efficiency and low memory requirement. So we select the Adam optimizer for this task. The revised contents are in lines 239-240.

Point 4: L. 260: replace section name with “Evaluation and Discussion”.

Response 4:

        Thank you very much for your suggestion to replace the section name. We have changed the section name according to the suggestion in lines 288, which is very helpful for this article.

Point 5: L. 282: “an astonishing performance” -> suggest to avoid such characterization

Response 5:

        Thank you for your comments. We apologize for the improper word. We have replaced the word “astonishing” with the “better” in the manuscript. The revised contents are in lines 310.

Point 6: 3.3. Experimental results: provide also the confusion matrix for the classification results.

Response 6:

        Thank you for this suggestion. We have added the confusion matrix of five classification results called Figure 5 in Experimental Results. The revised contents are in lines 280-284.

Figure 5. The confusion matrix of five validation images. Results in confusion matrix present percentage.

Point 7: Tables 2 and 3: supplement the results with ranking based comparison (provide mean rank of each method).

Response 7:

        Thank you for your advice. The results have been ranked on Overall Accuracy and Mean F1 score in Table 3 and Table 5. The bold font means the higher accuracy among these methods. As we can see, the DCCN is better than other methods both in Table 3 and Table 5, and the U-net is the worst. Moreover, we also provide the IoU and Mean IoU results to prove the effectiveness of these methods on measuring the shape and area of detected objects. The rank of effectiveness is shown as Table 4 and Table 6.

Point 8: Table 4: the improvement of your method DCCN over baseline is very small. Is it statistically significant? Perform statistical analysis over results to confirm or deny hypothesis about the statistical significance of improvement.

Response 8:

       Thank you for your sincere advice. The proposed method DCCN aims to make full use of multilevel features and decrease the loss of spatial information. So we have attempted several architectures by adjusting the location of coordconv module. The goal is to select the suitable architecture and achieve a better results. The results prove the DCCN is more suitable than other methods (as shown in Table 4). In addition, we also calculate the kappa coefficient of different architectures listed in Table 8. The mean kappa coefficient of DCCN is higher than other methods. This shows that the proposed DCCN is a suitable architecture and can achieve a better performance. The revised contents are in lines 349-351.

Table 8. The kappa coefficient results of the baseline, D2CCN and DCCN

Method

Image 1

Image 2

Image 3

Image 4

Image 5

Mean Kappa

baseline

84.52%

84.80%

82.25%

87.08%

83.57%

84.44%

D2CCN

85.32%

83.63%

80.39%

87.17%

82.31%

83.79%

DCCN

85.21%

84.72%

82.73%

87.32%

84.00%

84.80%

Point 9: Abstract: “significant improvement” -> a statistical proof is required or avoid using the word “significant”.

Response 9:

        We are sorry to use word improperly. We have added the classification accuracy of OA and Mean F1 score into the Abstract and adjusted the sentence without using “significant improvement”. The revised contents are in lines 27-28.

Point 10: L. 322-328: there is no Figure 9 in the article. Should be Figure 8.

Response 10:

        Thank you for your suggestion. We apologize for the wrong Figure 9. However, we have generated another Figure 5 in 3.3. Therefore the Figure 5-8 have been change into Figure 6-9.

Point 11: Conclusions: rewrite, be more specific, provide main numerical results (such as accuracy) from the proposed network. “the proposed model … was more robust” -> how did you evaluate robustness? Is it robustness to noise? I did not found robustness analysis in the article.

Response 11:

        Thank you for your advice. We have provided the classification accuracy of Overall accuracy and Mean F1 score in conclusion. The revised contents are in lines 393-394, 400-402. That is the “89.48% and 86.89% in Potsdam, 85.31% and 81.36% in Vaihingen” and “The OA and mean F1 score of proposed model are 1.36% and 1.16% higher than SegNet method in Potsdam, and far exceed the Deeplab-V3 and U-net methods”.

In addition, we are sorry about the improper content “the proposed model … was more robust” and we have removed it.

Round 2

Reviewer 1 Report

I am satisfied with the author’s answers to my concerns. 

Reviewer 2 Report

The authors worked well to revise the paper according to my comments and suggestions. The content of the paper has been improved significantly. I have no further comments and recommend the paper for acceptance.